# Modelling human liver fibrosis in the context of non-alcoholic steatohepatitis using a microphysiological system

Tomasz Kostrzewski [1✉], Sophie Snow[1], Anya Lindström Battle [1], Samantha Peel[2], Zahida Ahmad[3], Jayati Basak[3], Manasa Surakala [4], Aurelie Bornot[5], Julia Lindgren [6], Maria Ryaboshapkina [7], Maryam Clausen[6], Daniel Lindén[8,9], Christian Maass[10,11], Lucy May Young[1], Adam Corrigan[5], Lorna Ewart [3] & David Hughes[1]

Non-alcoholic steatohepatitis (NASH) is a common form of chronic liver disease characterised by lipid accumulation, infiltration of immune cells, hepatocellular ballooning, collagen deposition and liver fibrosis. There is a high unmet need to develop treatments for NASH. We have investigated how liver fibrosis and features of advanced clinical disease can be modelled using an in vitro microphysiological system (MPS). The NASH MPS model comprises a co-culture of primary human liver cells, which were cultured in a variety of conditions including+/− excess sugar, fat, exogenous TGFβ or LPS. The transcriptomic, inflammatory and fibrotic phenotype of the model was characterised and compared using a system biology approach to identify conditions that mimic more advanced clinical disease. The transcriptomic profile of the model was shown to closely correlate with the profile of patient samples and the model displayed a quantifiable fibrotic phenotype. The effects of Obeticholic acid and Elafibranor, were evaluated in the model, as wells as the effects of dietary intervention, with all able to significantly reduce inflammatory and fibrosis markers. Overall, we demonstrate how the MPS NASH model can be used to model different aspects of clinical NASH but importantly demonstrate its ability to model advanced disease with a quantifiable fibrosis phenotype.

[1] CN Bio Innovations Ltd, Cambridge, UK. [2] Discovery Biology, Discovery Sciences, R&D, AstraZeneca, Cambridge, UK. [3] Translational Biomarkers and Bioanalysis, Clinical Pharmacology and Safety Sciences, BioPharmaceuticals R&D, AstraZeneca, Cambridge, UK. [4] R&D IT, AstraZeneca, Chennai, India. [5] Quantitative Biology and Data Science, Discovery Sciences, R&D, AstraZeneca, Cambridge, UK. [6] Translational Genomics, Discovery Biology, Discovery Sciences, R&D, AstraZeneca, Gothenburg, Sweden. [7] Translational Science and Experimental Medicine, Research and Early Development Cardiovascular, Renal and Metabolism (CVRM), BioPharmaceuticals R&D, AstraZeneca, Gothenburg, Sweden. [8] Bioscience Metabolism, Research and Early Development Cardiovascular, Renal and Metabolism (CVRM), BioPharmaceuticals R&D, AstraZeneca, Gothenburg, Sweden. [9] Division of Endocrinology, Department of Neuroscience and Physiology, Sahlgrenska Academy, University of Gothenburg, Gothenburg, Sweden. [10] Certara, Quantitative Systems Pharmacology Unit, Sheffield, UK. [11] esqLABS GmbH, Saterland, Germany. ✉email: tomasz.kostrzewski@cn-bio.com

Non-alcoholic fatty liver disease (NAFLD) is a complex disorder that is rapidly growing in prevalence, especially in patients with type II diabetes and/or metabolic syndrome[1]. NAFLD includes non-alcoholic fatty liver (NAFL) and the severe form non-alcoholic steatohepatitis (NASH)[2]. A major hallmark of both conditions is the excessive accumulation of fat within the hepatocytes, known as macrovesicular steatosis. NASH is additionally characterised by a mixed inflammatory infiltrate and evidence of perisinusoidal/pericellular fibrosis[3] resulting from excessive collagen and extracellular matrix deposition by activated stellate cells, following injury to hepatocytes and/or activation of Kupffer cells[2]. The transition from NAFL to NASH is a dynamic, bidirectional process, strongly influenced by diet, physical activity, microbiota, environmental factors and genetic predisposition and NASH has become a leading cause of hepatocellular carcinoma (HCC)[2,4–7]. Disappointingly, despite the increasing global prevalence of NASH and burgeoning economic burden, there are no approved NASH therapies. This can be mainly attributed to inadequate knowledge of disease progression and its pathogenic drivers, particularly at the molecular level. However, several comprehensive research consortia now exist that are aimed at deepening the understanding of disease pathways, the discovery of specific biomarkers and the impact of co-morbidities or genotype and are giving hope that new innovative medicines are on the horizon.

The current in vitro and in vivo NASH models remain inadequate and consequently contribute to the lack of progress in understanding human disease aetiology and hamper the development of new medicines. It is widely acknowledged that no experimental model can fully recapitulate all aspects of the human disease, but that models with proven translational relevance, even if only for certain aspects of the disease, are of most value. Popular in vitro models include hepatic spheroids or precision-cut liver slices (PCLS). Spheroids are 3-dimensional cell models that facilitate cell-cell interaction and cell shape[8]. Hepatic spheroid models containing primary human hepatocytes (PHH)[9,10] or HepG2 cells plus hepatic stellate cells with the *PNPLA3* I148M sequence variant[11] or PHH with non-parenchymal cells[12], have been developed and have shown encouraging results with respect to recapitulating disease phenotypes and pharmacological modulation. PCLS models offer the advantage of retaining the multi-cellular architecture of the native liver and the presence of infiltrating immune cells, although viability in culture is limited to 5 days and hence only short-term exposure to free fatty acids (FFA) is possible[13,14]. On the other hand, in vivo models rely heavily on chemical or dietary-induced liver injury and/or on transgenics to produce lipid accumulation but not all the pathways are human-relevant and often the expression profiles do not mirror the human disease[15–17]. Hence, there remains a need to create models that faithfully recreate the human hepatic microenvironment to study mechanisms integral to disease onset, progression, and responses to new candidate drugs.

Microphysiological systems (MPS) show considerable promise for modelling aspects of NAFL[18] and NASH[19–21] and may alleviate some of the challenges encountered with the existing models. MPS models incorporate PHH and contain microfluidic technology which provides a dynamic perfusion throughout the duration of the experiment. The presence of this dynamic flow has been shown to enhance the period of time that PHH co-cultured with primary human Kupffer cells (KC), in liver microtissues, can be viably and functionally maintained[18,19,22,23]. In our recent publication, we extended the culture system to include primary human stellate cells (with and without the *PNPLA3* I148M sequence variant) and were able to maintain viability and functionality for up to 2 weeks[20]. Moreover, we showed that in the presence of FFA, liver microtissues accumulated intracellular triglycerides, produced pro-inflammatory

cytokines and early markers of fibrosis[20]. One key limitation of all current in vitro NASH models is their lack of robust fibrosis phenotype, which is essential to allow the model to be used to full evaluate the potential of new therapeutic interventions. Severity of fibrosis is fundamental to predict the risk of progression to cirrhosis and adverse clinical outcomes in NAFLD patients[24–26], and advanced fibrosis is particularly difficult to induce by metabolic factors in rodents[27].

Here we characterise the NASH phenotype in the tri-culture MPS model and investigate the biological cues driving a fibrotic disease state. An automated unbiased image analysis methodology enabled quantification of collagen deposition and α-smooth muscle actin (α-SMA) in liver MPS microtissues. We conducted a full transcriptomic and biomarker analysis of cells post-treatment with a range of cues representing the spectrum of potential pathogenic drivers. Profiles have then been compared to published datasets from human NASH patient samples. We identified conditions that represented the most advanced state of disease progression. Finally, we have extended the pharmacological characterisation of the model by analysing the effects of Obeticholic acid (OCA) and Elafibranor (ELF) as well as reversibility of the NASH phenotype upon removal of FFA from the cell culture media.

## Results

**Measurement of fibrosis in liver microtissues loaded with fat.** MPS models involving primary human hepatocytes exposed to FFA in culture media, containing physiological levels of glucose and insulin, have been shown to accumulate three times more fat compared to hepatocytes not exposed to FFA whilst maintaining their viability and functionality over a 2-week period[18]. In this way, features of NAFL are recapitulated and herein we refer to these microtissues as steatotic. But in vivo the liver is a multicellular organ and crucially the presence of Kupffer cells and hepatic stellate cells (HSC) have been shown to play a pivotal role in the progression of NAFL into NASH[2,21,28,29]. Indeed, one of the distinguishing histological features of NASH is the presence of fibrosis owing primarily to the activation of HSC, which causes increased collagen-1 deposition. However, in published MPS studies collagen-1 has not been directly measured, instead expression of surrogate markers such as secreted pro-collagen 1, TGFβ and osteopontin are reported[19,20], in addition to gene expression changes[19,20]. Here, we have developed a high content, automated imaging application that enables quantification of collagen-1 deposition and spatial localisation of α-SMA expression within HSC, an indicator that quiescent HSC has dedifferentiated into motile myofibroblasts (Supp. Fig. 1). Compared to control and steatotic microtissues (PHH with/without FFA exposure) there was a sixfold increase in α-SMA expression and an eightfold increase in collagen-1 deposition in NASH microtissues (Fig. 1a, b). These changes were corroborated by measurement of secreted clinical fibrotic markers TIMP-1, pro-collagen 1, YKL-40 and fibronectin, which were all significantly increased in NASH compared to steatotic and control microtissues (Fig. 1c) and the increased expression of genes associated with fibrosis in the NASH microtissues (Fig. 1d and Supp. Table 1). During the progression of NAFL to NASH, there is an increase in production of pro-inflammatory cytokines, most likely initiated by activated KC. As previously reported our NASH microtissues produced significantly higher levels of a wide variety of inflammatory cytokines associated with NASH including IL-1β, IL-6, TNFα, MCP-1 and MIP-1β (Supp. Fig. 2a), without affecting the cell health assessed by tissue density of the microtissues and albumin production (Supp. Fig. 2b, c). Fat loading in the NASH microtissues was confirmed using Oil Red O staining (Supp. Fig. 2d, e). We additionally established that the phenotypic

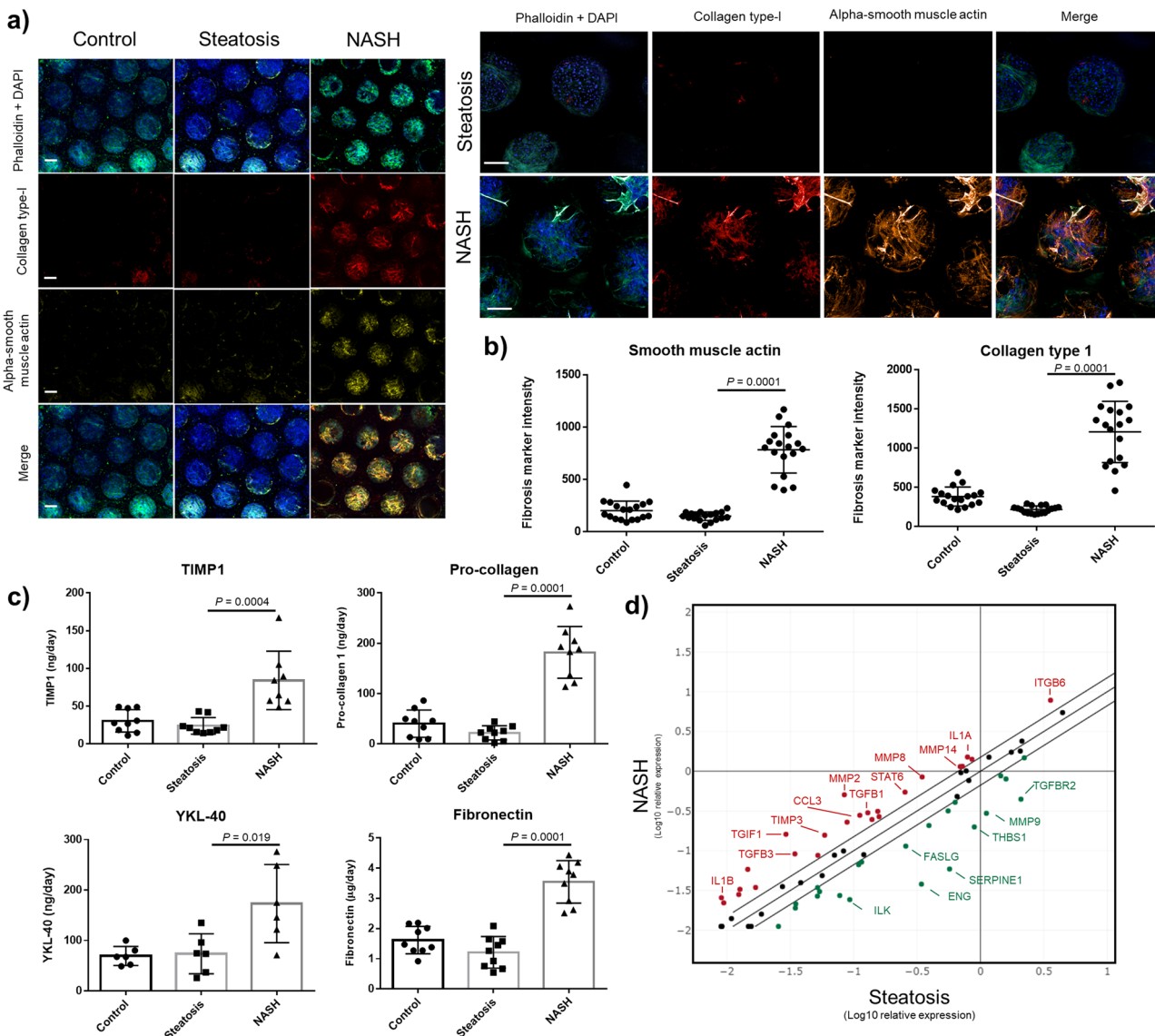

**Fig. 1 Liver MPS NASH model demonstrates fibrotic phenotype.** PHH alone (Control and Steatosis) or PHH, KC and HSC co-cultures (NASH) were cultured in the MPS platform for 14 days under standard media (control) or high fat conditions (Steatosis and NASH). **a** Liver microtissues were stained for cytoskeleton (phalloidin), collagen-type I and α-SMA and imaged by confocal microscopy. Representative images shown and scale bars 200 μm. **b** Staining of microtissue was quantified by measuring total fluorescence intensity throughout individual microtissues, each data point represents an average of all microtissues within a FOV (min 8, max 12) and two FOV per scaffold. **c** Secreted fibrosis biomarkers TIMP-1, Pro-collagen 1, YKL-40 and Fibronectin were all measured in cell culture medium by ELISA, at the end of the culture. **d** The expression of fibrosis genes was analysed in total RNA from steatosis and NASH models using Human Fibrosis RT2 Profiler PCR Arrays. Gene expression levels are expressed as Log10 relative expression compared to housekeeping genes (GAPDH/B2M/HPRT1) and compared between the NASH and steatosis models (red = upregulation, green = downregulation, black = no change, upper line—1.5-fold increase, lower line 1.5-fold decrease). All data points shown with means ± SD highlighted, data generated from a minimum of nine independent cultures (three donors per condition and $n = 3$ per donor).

changes observed in the NASH microtissues were not affected by the biological donors used, particularly as non-donor matched material was always used to generate NASH microtissues (Supp. Fig 3). The same pattern of expression was observed, with some variation in absolute values, for a wide variety of cytokines and biomarkers when comparing NASH microtissues produced from three separate donors of PHH, KC and HSC (Supp. Fig 3).

**Transcriptomic signature of NASH microtissues aligns with human data and is significantly different to that from commonly used rodent models of the disease.** The molecular events that drive NASH are poorly understood, but the transcriptomic signatures derived from NASH livers offer an opportunity to identify molecular pathways and their contribution to NASH pathogenesis. Using RNAseq, we conducted transcriptomic profiling of our MPS model and transcriptomic profiles of control lean microtissues mapped to non-diseased human liver profile (Supp. Fig. 4a). The control MPS retained expression of PHH, HSC and KC markers (Supp. Fig. 4b–d) providing confidence that the cells in the microtissues together with the micro-environment created by the MPS device combine to give a relevant model system to study disease. Comparing the profile of NASH microtissues (PHH, KC, HSC with FFA) with the Human Gene Atlas database (www.ebi.ac.uk/gxa/home) identified a human liver profile (Fig. 2a) and comparison to the DISEASES

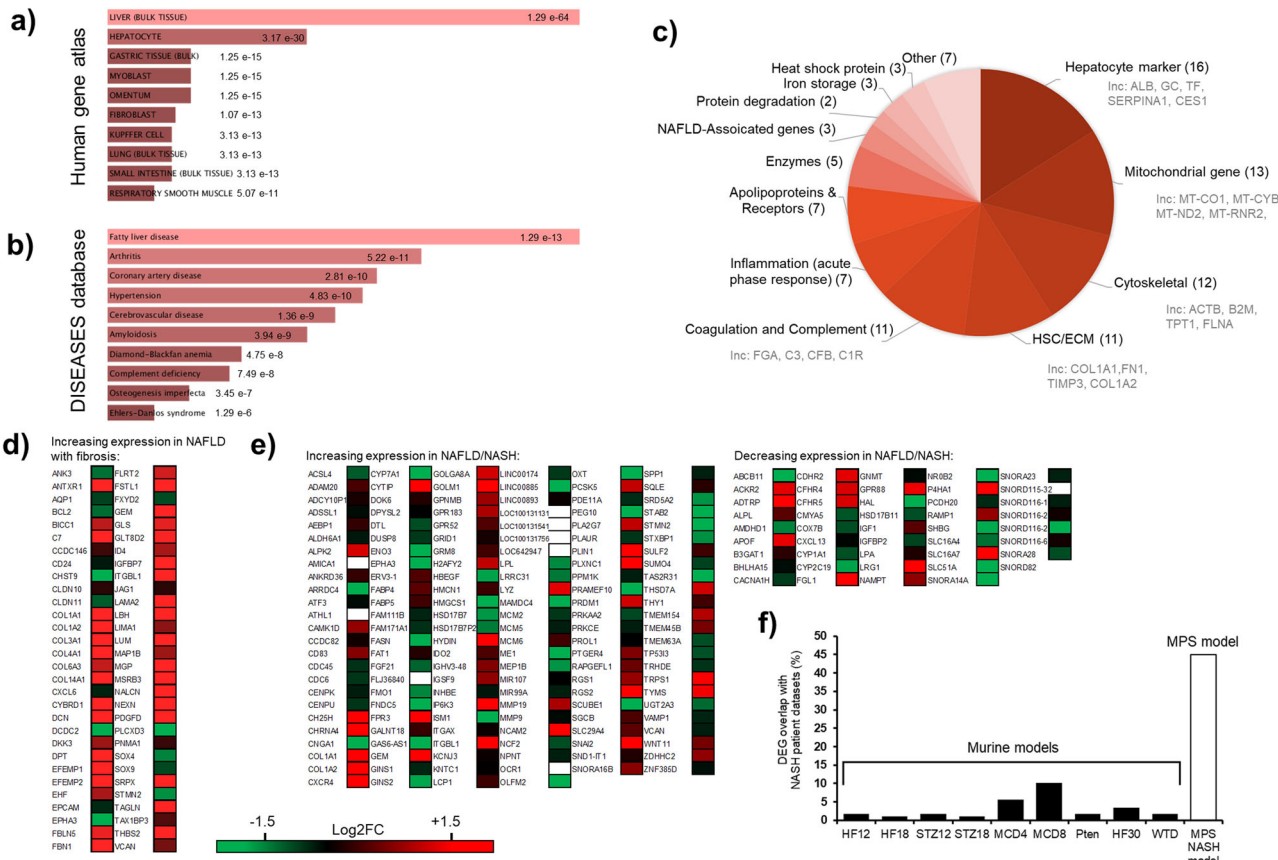

**Fig. 2 Transcriptional profile of NASH liver MPS model aligns with profile from human clinical samples.** PHH, KC and HSC co-cultures were cultured in the MPS platform under high fat conditions for 14 days and total RNA extracted and analysed by RNAseq to determine overall transcriptomic profile. Transcriptional profile of liver NASH MPS compared to the **a** Human gene atlas and **b** the DISEASES database using the Enricher gene enrichment analysis tool (maayanlab.cloud/Enrichr/)[68]. 300 highest expressed DEGs in the NASH model were compared to control PHH samples, ranking by *p* value (probability of genes being associated with the gene set. **c** Breakdown of Top 100 genes expressed in liver MPS NASH model. **d**, **e** Published studies identify key differentially expressed genes specifically in NAFLD/NASH patient samples. **d** Advanced vs mild fibrosis in NAFLD[30], **e** NASH vs NAFLD vs obese control[17]. Heatmaps demonstrate Log2 fold change in transcripts for these genes in transcriptome from liver MPS NASH model compared to control PHH samples. **f** Comparison of correctly overlapping DEGs in NASH pre-clinical models, murine models from literature[17] and NASH MPS model (data from **e**). Transcriptomic data were generated from nine independent replicate samples and averaged.

database (https://diseases.jensenlab.org/Search) produced a close correlation with human fatty liver disease (Fig. 2b). These results provide confidence that the microtissues are exhibiting good similarity to the disease.

The breakdown of the top 100 genes expressed in the NASH microtissues (Supp. Table 2) indicated that the vast majority (77%) of genes were associated with hepatic function, metabolism, inflammation, and fibrosis (Fig. 2c). To further demonstrate the translational relevance of the MPS NASH model we compared the transcriptional profile to gene lists from literature[17,30] that are differentially expressed in human NASH patients (Fig. 2d, e). For the dataset from Moylan et al.[30], of all the differentially expressed genes (DEGS) from NASH patients, 75% were also differentially expressed in the NASH MPS microtissues (Fig. 2d). In the study by Teufel et al.[17], 193 genes were identified as representing key changes in human NAFLD/ NASH and these genes were then compared to nine standard mouse pre-clinical NASH models. The study demonstrated between 0.01% and 10% of the DEGs in NASH patients were also differentially expressed in the mouse models indicating their low human relevance. Comparison to the transcriptomic signature of our NASH microtissues indicated a far greater relevance to the patient samples with up to 45% of the DEGs being correctly differentially expressed (Fig. 2e, f).

**Reversibility of disease phenotype and pharmacological modulation in NASH liver microtissues loaded with fat**. Despite many on-going clinical trials, there is still no approved treatment for NASH, but emerging data provide confidence that NASH is a pharmacologically responsive disease. As such, the drug discovery process would significantly benefit from easy-to-use, human-relevant models. Having demonstrated that our NASH MPS microtissues are representative of the human disease, we next assessed its pharmacological responsiveness to two agents in late-stage clinical development, OCA and ELF. First, we set out to determine an optimal dosing strategy for each of the treatments based on an assessment of their physicochemical properties and pharmacokinetic behaviour within the MPS device (Table 1). Non-specific binding to the MPS device was determined after a 72 h incubation with each candidate drug at 1 μM (Table 1). Both candidate drugs were calculated to have less than 0.1% loss as a consequence of non-specific binding to the MPS device (Table 1). Protein binding of each candidate drug within the fat culture medium was also measured and we found that OCA and ELF were highly protein-bound, highlighting that higher concentrations of the drugs would need to be added to the microtissues to ensure sufficient free drug availability for pharmacological effect (Table 1). The protein binding calculation for OCA is comparable to reported plasma protein binding[31], but an equivalent figure for

**Table 1 Determining translationally relevant drug concentrations for use in liver MPS NASH model studies.**

|  | Obeticholic acid | Elafibranor |
|---|---|---|
| Drug recovery after 48 h incubation in blank MPS platform (no cells) | 110% (±5.5%) | 97.0% (±7.6%) |
| Protein binding in fat cell culture media | >99% (N/A) | 98% (±0.1%) |
| $T_{1/2}$ in MPS platform | 12 h (±5.5%) | 16 h (±7.5%) |
| Clinical dose | 10 mg or 25 mg dose QD | 80 mg or 120 mg QD |
| Plasma Cmax (and liver Cmax) | 0.5 µM (12.5 µM) | 13 µM (not known) |
| Plasma protein binding | >99% | Not available |
| Proposed dosing strategy for MPS model | Dosed daily into OOC at 0.5 µM, 5.0 µM and 12.5 µM | Dosed daily into OOC at 1.3, 13 and 65 µM |

To determine non-specific compound binding of Obeticholic acid (OCA) and Elafibranor (ELF), both compounds were incubated in liver MPS plates for 48 h and compound concentration in cell culture media was compared to input media by quantitative LC-MS. Rapid-equilibrium dialysis was used to determine protein binding of the compounds in HEP-FAT media. The $T_{1/2}$ of both compounds was determined by incubating 1 µM of compound with NASH liver microtissues grown for 7 days in liver MPS plates. The concentration of compound in cell culture medium was measured across 48 h by LC-MS. Data for these studies were generated from a minimum of three independent cultures per condition. These data for both compounds were compared to literature values for plasma Cmax concentrations[31,58,70], liver Cmax concentrations[31] and plasma protein binding[31].

ELF was not available. Because the hepatocytes within the MPS device are metabolically competent, we evaluated the half-life for each candidate drug in NASH microtissues. OCA was found to have a half-life of 12 h, whereas ELF had a half-life of 16 h (Supp. Fig. 5). Using these data, dosing strategies for both compounds were identified which maintain drug concentrations close to clinically relevant levels throughout the treatment period (Table 1).

Following 10 days of compound treatment (Supp. Fig. 6a), NASH microtissues were assessed for the production of inflammatory and fibrotic biomarkers (Fig. 3a), the expression of pro-fibrotic genes (Fig. 3b) and the expression of collagen-1 and alpha SMA (Fig. 3c, d). Both OCA and ELF caused a concentration-dependent decrease in inflammatory cytokines, including: CXCL1, IL1-Rα, IL-6 and MCP-1 (Fig. 3a). Both compounds also reduced expression of genes associated with ECM remodelling (e.g. MMP1, MMP9 and SERPINA1), TGFβ signalling (CEBPB, DCN) and inflammation (CCL2, CCL3, CXCR4) (Fig. 3b). Neither compound significantly affected cell health or microtissue formation, but higher doses of ELF did slightly reduce albumin production (Supp. Fig. 6b, c). Crucially using the developed quantitative imaging approach, both OCA and ELF abrogated α-SMA expression in a concentration-dependent manner by up to 50% and collagen-1 deposition was reduced by up to 35% (Fig. 3d).

Bariatric surgery can significantly reduce NASH and liver fibrosis in patients with obesity, as described in recent meta-analyses[32,33], so in addition to pharmacological intervention we also explored whether the disease phenotype in the NASH microtissues could be reversed by reducing the FFA in the culture medium, as a mimic of dietary changes in humans. NASH microtissues were cultured in lean or fat conditions for a total of 30 days, with some microtissues cultured in fat medium for the first 15 days and then switched to lean medium for the remaining time. At day 15 and at day 30 the phenotype of the microtissues was assessed. By switching the microtissues from fat to lean media inflammation (IL-6 and MCP-1 production) was reduced (Fig. 3e, f), markers of fibrosis (TIMP1) were reduced (Fig. 3g) and fat loading was halted (Fig. 3h). Together these data demonstrate that the NASH microtissues are well placed to assess the efficacy of a range of therapeutic interventions and assess the full spectrum of disease endpoints from fat loading/steatosis through to liver fibrosis.

**Systems biology interrogation of NASH microtissues demonstrates a spectrum of disease phenotypes can be generated.** There is only limited understanding of the pathogenic drivers of

NASH and their interaction, principally due to a lack of translational models that are amenable to detailed molecular analysis[2,34]. Here, a systems biology 'cue-signal-analysis' paradigm[35,36] was utilised to produce large factorial datasets to enable the identification of molecular pathways of importance. We first compared monocultures of PHH with co-cultures with and without both KC and HSC to determine which combination gave the phenotype most akin to the human disease and observed that all NASH biomarkers were only produced with all three cell types present in the microtissues (Supp. Fig 7). To create varying set ups of the NASH microtissues we created hepatocyte microtissues in the MPS with high numbers of NPC (60,000 KC and HSC—equivalent to 10% of microtissue population) and low numbers of NPCs (6000 KC and HSC—equivalent to 1% of microtissue), representing physiological and non-physiological cell ratios respectively. Next, we took these different microtissues and cultured them in media with or without FFA. Finally, we supplemented the media with lipopolysaccharide (LPS), TGFβ, fructose or cholesterol either as single 'cues', in pairs or all together (Fig. 4a). 'Cues' were selected for clinical association with NASH. LPS, is a pleiotropic inflammatory molecule, it is proposed to reach the liver via the hepatic portal vein after release from the gastrointestinal tract, contributing to disease progression[37,38]. TGFβ has been extensively studied in relation to fibrosis due to its regulation of genes controlling collagen and matrix-modifying enzymes[39]. Fructose is a key substrate for de-novo lipogenesis, a process that has been identified as a key event in lipid accumulation within NAFLD patients[40]. Finally, cholesterol accumulation has also been linked to NASH disease aetiology[41].

We compared the transcriptional profile of the NASH MPS model with and without each cue to determine how they affected the disease state. Using GO term enrichment and cluster analysis indicated that the cues affected processes implicated in the multi-hit hypothesis of NASH pathogenesis[42]: extracellular matrix organisation, mitochondrial function, inflammation and responses to stress (Fig. 4b). These effects and clusters were substantially much more prominent in the high NPC condition than the low NPC condition (Supp. Fig. 8). LPS only induced pathways associated with inflammatory signalling, not all of which are relevant to NASH, whilst TGFβ and combinations of cues including TGFβ induced a wide range of clusters associated with collagen formation and matrix reorganisation (Fig. 4b).

We also assessed translational relevance of the microtissue model by comparison of differential expression changes to marker gene sets derived in NAFLD patient cohorts. We firstly determined if the microtissues recapitulated the molecular signatures of processes that occur early in the development of

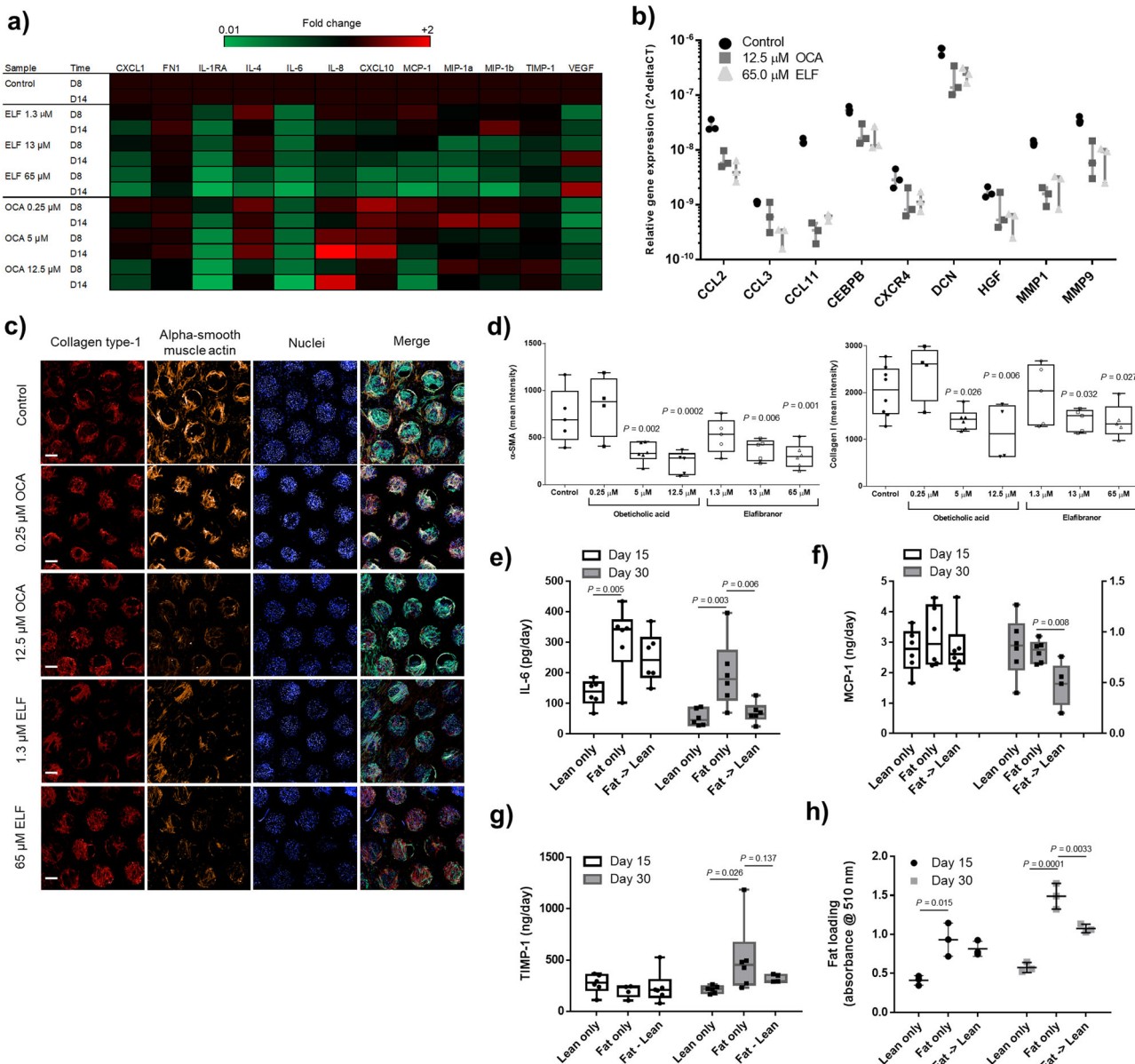

**Fig. 3 Obeticholic acid and Elafibranor both modulate inflammatory and fibrosis phenotype in liver MPS NASH model and phenotype can additionally be reversed with dietary changes.** PHH, KC and HSC co-cultures were cultured in the MPS platform under high fat conditions and dosed with varying concentrations of Obeticholic acid (OCA) and Elafibranor (ELF) QD or vehicle control (control) for 10 days, following an initial 4-day pre-culture phase. **a** The secreted cytokine profile from liver microtissues was compared by Luminex to determine the effects of each compound on the inflammatory profile of the liver with samples analysed at day 8 and 14 of the culture. Data were normalised by Z-transformation to allow comparison of all analytes and presented as a heatmap. **b** The expression of fibrosis-associated genes was analysed in total RNA from control, OCA and ELF treated liver microtissues (highest concentration for each compound) using Human Fibrosis RT2 Profiler PCR Arrays. Gene expression changes are expressed as a fold change over control. **c** Liver microtissues were stained for collagen-type I, α-SMA, nuclei (DAPI - blue), phalloidin (green) and imaged by confocal microscopy. Representative images are shown and scale bars 200 μm. **d** Staining of microtissue was quantified by measuring total fluorescence intensity throughout individual microtissues, each data point represents an average of all the microtissues imaged within an MPS scaffold (min 8, max 20 microtissues). **e** PHH, KC and HSC co-cultures were cultured in the MPS platform under high fat or lean conditions for 30 days. Cultures either remained in the same type of media throughout or were switched at day 15 from fat media to lean media (Fat -> Lean). Cell culture medium samples were analysed for the presence of IL-6, **f** MCP-1, **g** TIMP-1, which were measured by ELISA. **h** Microtissues were stained with Oil-Red O to determine fat loading during culture period, total stain was quantified by absorbance at 510 nm. All datapoints shown, either as box-whisker plots highlighting mean and min-max or with error bars highlighting means ± SD. All data from a minimum of three independent cultures; statistical comparisons made to control samples unless other comparison shown.

NAFLD in humans, namely lipid accumulation, insulin resistance, mitochondrial dysfunction and changes to the poly (ADP-ribose) polymerases (PARP) pathway. FFA and the combination of FFA plus cholesterol, but not the other single cues, affected lipid droplet-associated genes[43] in a manner resembling simple steatosis in NAFLD (Supp. Fig. 9). Fructose and fat (FFA) challenge combined with fructose-induced differential expression changes consistent with insulin resistance[44] (Supp. Fig. 10). Most cues also reduced the expression of mitochondrial associated genes (Supp. Fig. 11) and challenge with FFA was sufficient to

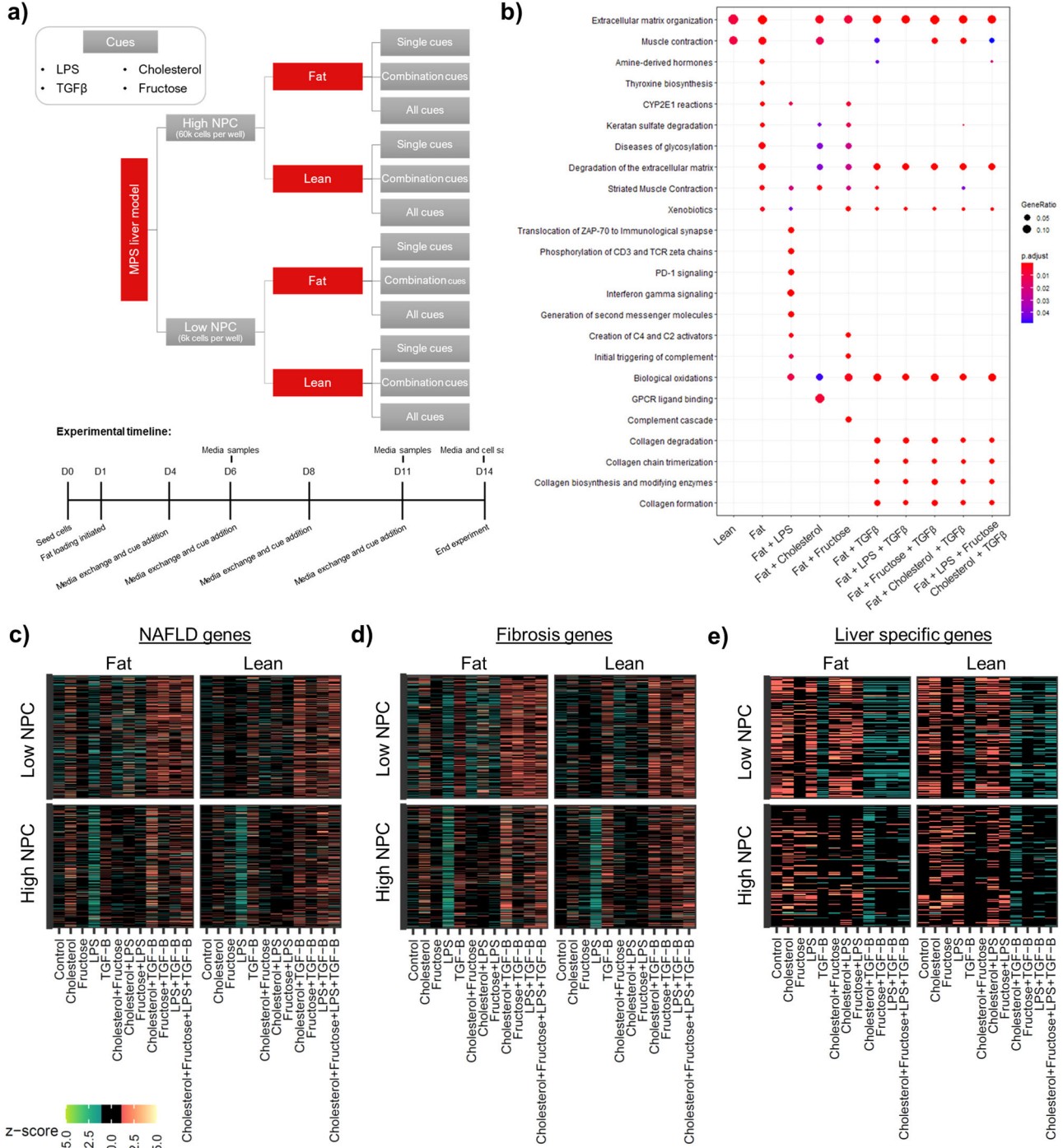

**Fig. 4 Systems biology interrogation of liver MPS NASH model to determine physiologically relevant cues that promote transcriptional changes to mimic advanced NASH disease state.** PHH, KC and HSC co-cultures were cultured in the MPS platform under a variety of conditions for 14 days and transcriptomic profile of microtissues were compared. **a** Study was designed to test number of NPCs, presence of fat, fructose, cholesterol, LPS and TGFβ for effects on the transcriptional profile of the liver MPS NASH model. **b** Differentially expressed genes (defined by comparison to low NPC lean) from selected conditions were compared to identify enriched pathways and biological processes, using PANTHER database and mapped using CompareCluster, with size of cluster representing number of genes involved (GeneRatio) and colour identifying confidence interval (p.adjust). Only high NPC conditions shown (low NPC data in Supp. Fig. 8) and some conditions did not produce any gene clusters with significant *p* value < 0.05 and are not shown on plot. **c** Expression of markers of NAFLD activity score progression[45]. **d** Expression of markers of fibrosis stage progression in NAFLD[45]. **e** Expression of liver-specific genes[46]. Heat maps used to compare different culture conditions with data normalised to z-scores to show variance from mean of all observations for each gene. Transcriptomic data were generated from a minimum of three independent replicate samples per condition.

modulate PARP pathway, but changes were moderate and additional cues did not cause further changes (Supp. Fig. 12). Interestingly, there were differential gene expression changes with the expected directionality to NAFLD in the microtissues with the lower number of HSCs and KCs (6 K) as well as the microtissues with a more physiological number (60k) of non-parenchymal cells (Supp. Figs. 9, 10) and the changes in mitochondrial genes were most pronounced in the low NPC group (Supp. Fig. 11).

Next, we assessed if the NASH microtissues re-capitulated molecular features of disease progression. TGFβ and combination of cues with TGFβ mirrored a gene signature for NAFLD progression[45] (Fig. 4c) and fibrosis stage progression[45] (Fig. 4d) and a reduced signature for liver-specific genes[46] (Fig. 4E), reflecting decline in normal liver function in advanced disease. Confirmatory analyses indicated that TGFβ and, to a lesser extent, fructose triggered gene expression changes resembling fibrosis[47] (Supp. Fig. 13), overexpression of stellate cell signature[48] (Supp. Fig. 14) as well as differential expression changes of markers associated with HCC risk[49,50] (Supp. Fig 15). In particular, genes associated with collagen (e.g. COL3A1) whose deposition is associated NASH progression[47] were elevated (Supp. Fig. 13). TGFβ and cues combined with TGFβ reduced expression of core metabolism and transporter genes[51] (Supp. Fig. 16), mimicking reduced detoxication ability of NASH livers[52]. Treatment with LPS did not mimic transcriptomics profiles of human NASH or NAFLD.

Assessment of gene expression changes was supplemented by assessment of the secretome produced by NASH microtissues in the MPS after exposure to the various cues. Increased pro-inflammatory cytokine secretion is a defining characteristic in NASH patients, so we conducted a Luminex cytokine array covering 42 different biomarkers, cytokines and chemokines secreted into the culture media of the different microtissues at multiple time points. We consistently detected 27 out of the 42 biomarkers and observed that there was differential expression of cytokines and chemokines driven by the different cues applied to the micro-tissue (Fig. 5a, b). The LPS cue and the presence of higher NPC numbers were responsible for driving the largest increase in cytokine and chemokine release, with the largest increases in cytokines measured on day 14 (Fig. 5b). Specifically, for LPS dosed samples IL-4, GM-CSF and MIP1α showed the biggest increase compared to control. IL-6 and IL-8 also have recognised chemoattractant properties and are up-regulated in NAFL/NASH patients[2] but interestingly, although LPS caused an increase in their secretion, it was not as high as IL-4, GM-CSF and MIP1α (Fig. 5b). Fructose caused the biggest increase in IP-10 production, occurring on day 15 (Fig. 5b). The greatest increase in TNFα also occurred on day 14 in response to a cholesterol cue (Fig. 5b). However, the data also indicated that there was inter-cue crosstalk with respect to cytokine production, with most cues in combination resulting in a suppression of inflammation markers (Supp. Fig. 17). For the pro-fibrotic biomarkers, again the increased number of NPCs, TGFβ and to a lesser extent fructose caused the most significant increases in pro-collagen 1, fibronectin and TIMP-1 expression (Fig. 5c). Similarly, to the cytokine production, these changes were not further enhanced by combining cues, as all the cue combinations containing TGFβ had the highest expression (Supp. Fig. 17c). Albumin production was similar between lean and fat microtissues irrespective of what cue was studied, but the presence of TGFβ and to a lesser extent cholesterol reduced albumin production (Supp. Fig. 18), an observation which mirrors the transcriptomic data suggesting TGFβ in particular causes a decline in liver function as expected in more advanced NASH[52].

By using this systems biology 'cue-signal-analysis' paradigm to interrogate the MPS NASH model we have identified that the MPS NASH model can be created with varying levels of NPCs but with higher and more physiologically relevant NPC numbers are all soluble biomarkers detectable. FFA and the combination of FFA with cholesterol can be used to model steatosis. Treatment of the MPS NASH model with TGFβ and to a lesser extent fructose, could be used to model inflammation and fibrosis in NASH and potentially study the development of NASH-induced hepatocellular carcinoma, because TGFβ may be pro-carcinogenic[50].

## Pharmacological modulation of NASH microtissues with enhanced fibrosis.

The systems biology approach identified TGFβ to be a key driver in creating a more advanced NASH phenotype in the MPS model, with enhanced fibrotic markers and reduced liver function. Transcriptomic effects were observed with both high and low NPC groups, so we compared the fibrosis by microscopy of NASH microtissues with varying numbers of NPCs in the presence of fat and TGFβ (Supp. Fig. 19). Microtissues with enhanced NPC numbers had significantly higher levels of collagen-I and alpha SMA expression and were more responsive to TGFβ (Supp. Fig. 19). Therefore, we explored how effective pharmacological intervention would be for NASH microtissues with high NPC numbers supplemented with TGFβ and exposed them to varying concentrations of OCA and ELF QD as previously described (Supp. Fig. 6a). Similarly, to previous observations, both candidate drugs were shown to have profound inhibitory effects on inflammatory cytokine production (Fig. 6a). OCA had the greatest effect on IL-1Rα whereas ELF had the greatest effect on MCP-1 reflecting the fact that while both candidate drugs have anti-inflammatory properties, but their mechanisms of action are different. The differences in the compounds were further illustrated when exploring effects on the soluble biomarkers TIMP-1 and pro-collagen 1, both of which were reduced by ELF treatment but not OCA treatment (Fig. 6b). Finally, we used our quantitative fibrosis imaging assay to assess the effect of each candidate drug on collagen-1 deposition and α-SMA expression (Fig. 6c). We observed a significant increase in collagen-1 and α-SMA signal in NASH microtissues treated with fat and TGFβ compared to just fat alone (Fig. 6c). Both candidate drugs caused a significant reduction in collagen-1 deposition and α-SMA expression, with the highest concentrations of OCA and ELF causing an 80% reduction in α-SMA expression and a 75% reduction in collagen-1 deposition (Fig. 6c). These reductions are more significant than those previously observed in the fat only NASH model (Fig. 3) and demonstrates the value of the MPS NASH model to be able to investigate the effects of therapeutic intervention on varying stages of disease from simple steatosis, to NAFLD and through to advanced NASH with significant fibrosis.

## Discussion

Here we evaluated PHH microtissues co-cultured with human HSC and KC in an MPS which provides continual dynamic perfusion of cell culture media through the microtissues. We demonstrate that in the presence of FFA together with physiological levels of glucose and insulin the microtissues exhibit transcriptomic and cytokine profiles consistent with the human NAFL/NASH disease phenotype. Moreover, these microtissues exhibit key features of fibrosis such as collagen-1 deposition and increased α-SMA expression, which are both quantifiable using automated confocal imaging and are pharmacologically responsive to the farnesoid X receptor (FXR) agonist OCA or the dual PPARα/δ agonist ELF. Extending the model characterisation further, we have examined the responsiveness of the microtissues to six biological cues either independently or in combination. Each cue was chosen to represent a different pathogenic driver thought to be implicated in the transition of NAFL to NASH. We

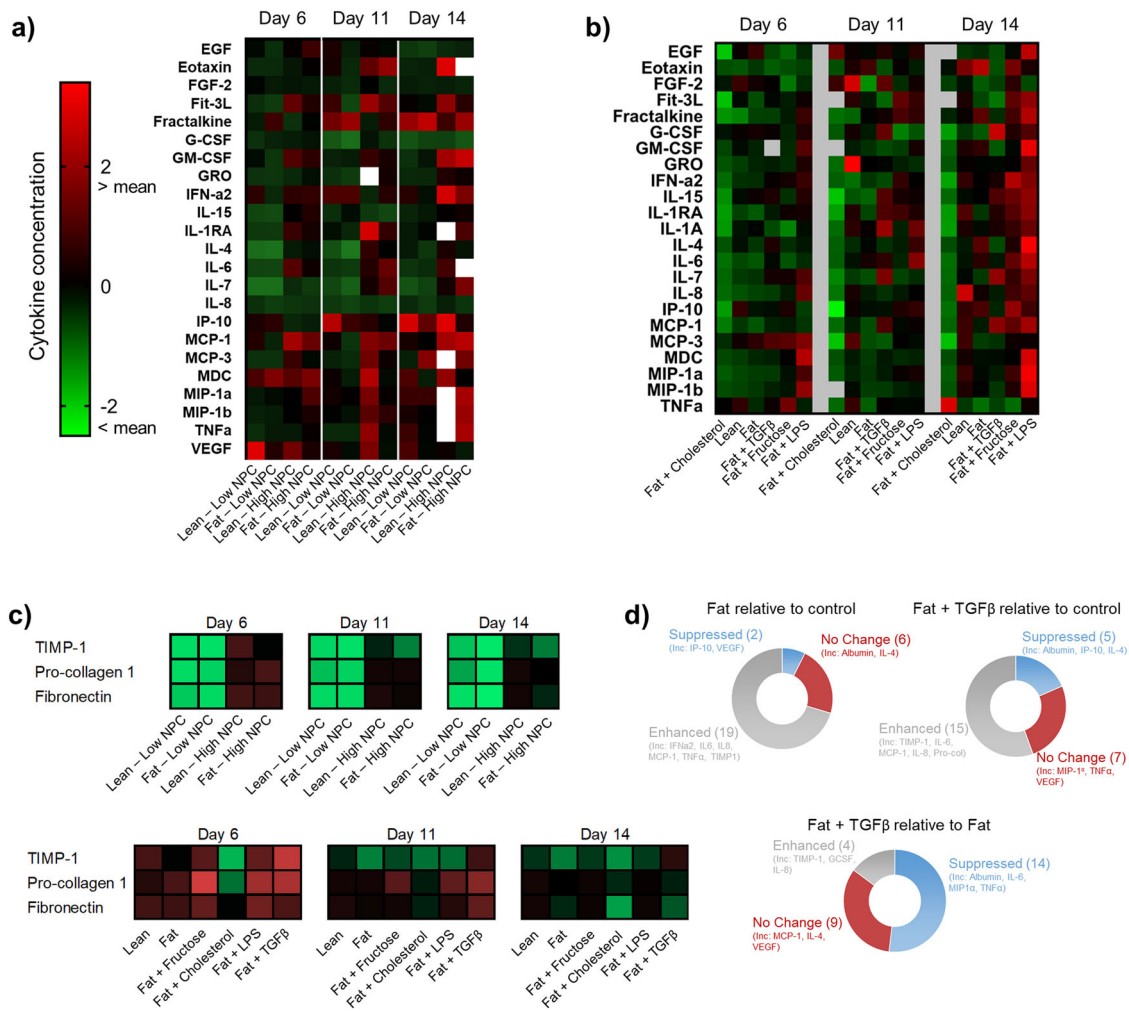

**Fig. 5 Systems biology interrogation of liver MPS NASH model to determine physiologically relevant cues that promote soluble biomarker changes to mimic advanced NASH disease state.** PHH, KC and HSC co-cultures were cultured in the MPS platform under a variety of conditions for 14 days to test number of NPCs, presence of fat, fructose, cholesterol, LPS and TGFβ for effects on the soluble biomarker profile of the liver MPS NASH model. Expression of biomarkers, cytokines and chemokines were compared, all concentrations were normalised by Z-transformation to allow comparison and shown relative to mean for each marker (white = not detected). **a** Cytokine expression compared with varying levels of NPCs; **b** for the high NPC group in the presence of additional cues; **c** Expression of pro-fibrotic biomarkers with varying NPC content and with additional varying cues in high NPC condition (additional data in Supp. Fig. 17). **d** Graphical summary of the changes that occur in the NASH model in the presence of Fat and Fat + TGFβ. All data were generated from a minimum of three independent replicate samples per condition.

found that exposure to FFA, TGFβ and fructose provided a sound model of inflammation and fibrosis in NASH whereas FFA alone or in combination with cholesterol was a suitable model of steatosis and the earlier stages of NASH. Despite the already high prevalence of NASH there is no approved drug for treatment and furthermore, there is little consensus on the pathogenic drivers of disease, probably due to the poorly representative disease models. Here we have introduced a well-characterised human in vitro model that exhibits key histopathological features of disease and together with the composite array of measurable endpoints that this model is suitable for investigating disease pathways associated with NASH pathogenesis as well as for employment within drug discovery programs.

Essential to this model's utility in drug discovery and extending biological understanding of NASH is its ability to recapitulate major aspects of human disease such as fat accumulation, pro-inflammatory cytokine production and hallmarks of fibrosis such as increased gene expression relating to collagen production, deposition of collagen-1 and increased expression of α-SMA within HSC. In addition, the microtissues were incubated

throughout all experimental procedures in cell culture media that contained physiological levels of glucose and insulin, significantly lower than standard hepatocyte cell culture medium (typically glucose = 11 mM and insulin = 1.1 μM), allowing us to test the consequences of adding FFA as well as pathogenic cues such as LPS, fructose, cholesterol or TGFβ. A distinguishing aspect of this investigation is that considerable attention has been paid to recapitulating the liver microenvironment through the application of fluid flow (perfusion) as well as multi-cellular cultures in ratios that were deemed physiological. Consequently, the hepatocytes and accompanying non-parenchymal cells retain their phenotype and function, extending their viability for at least 4 weeks in culture as previously observed[18,20,22]. This longevity of function allows chronic/repeat drug dosing studies to be performed, more equivalent to in vivo studies, as well as studies to investigate the impact of dietary/media components. This combination of features demonstrates the MPS model has a number of advantages over other in vitro liver disease models. For example, precision-cut liver slices of disease tissue are able to maintain in vivo histopathology, but these models only typically

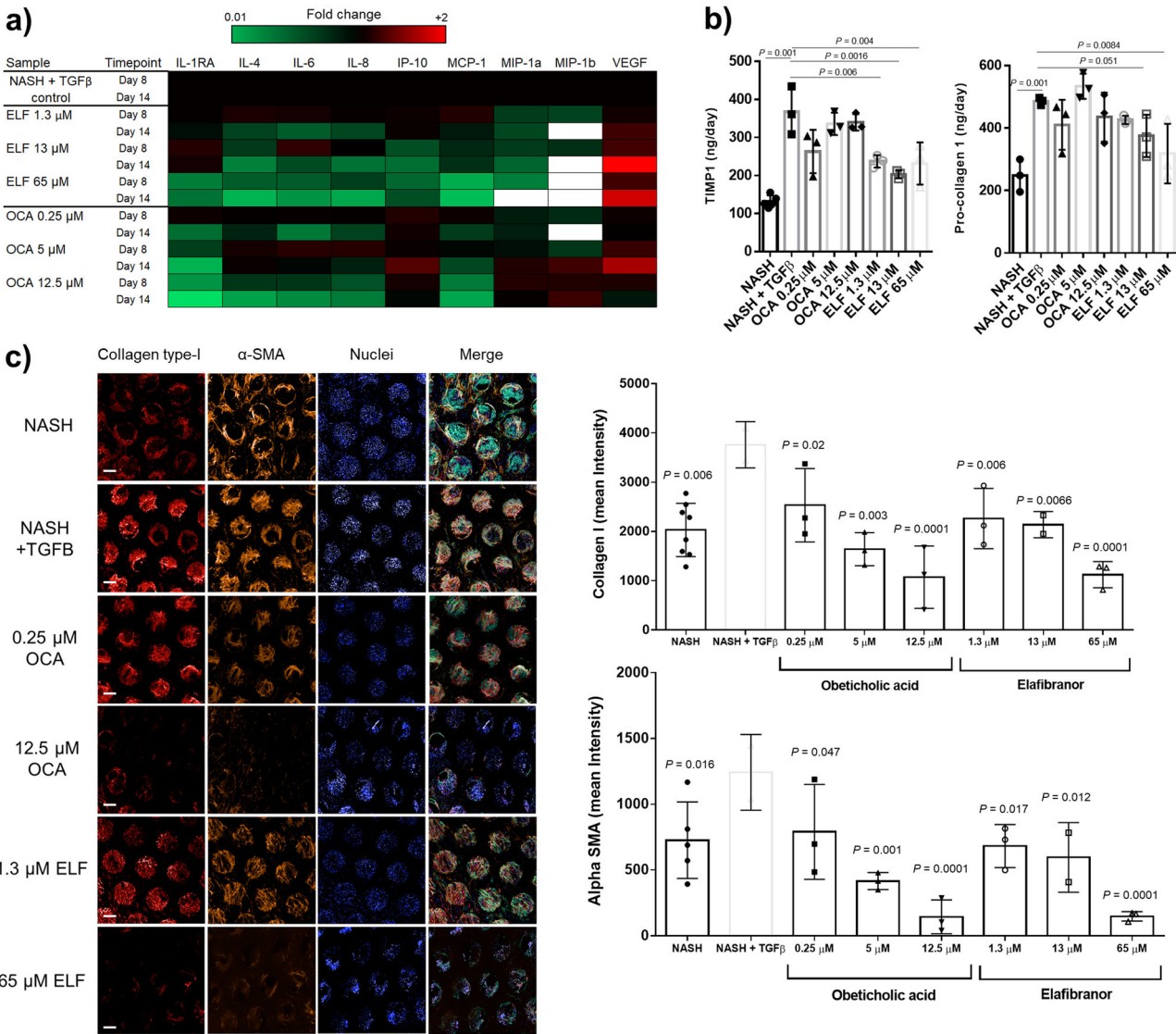

**Fig. 6 Using enhanced liver MPS NASH model with TGFβ, Obeticholic acid and Elafibranor modulate disease phenotype to a greater extent than in standard NASH model.** PHH, KC and HSC co-cultures were cultured in the MPS platform under high-fat conditions and dosed with varying concentrations of Obeticholic acid (OCA) and Elafibranor (ELF) QD or vehicle control for 10 days, following an initial 4-day pre-culture phase. Throughout the compound dosing period TGFβ was also dosed onto all microtissues (NASH only vehicle control samples were also included in the study for comparison, which were not treated with TGFβ or dosed with any compound). **a** The secreted cytokine profile from treated liver microtissues was compared to determine the effects of each compound on the inflammatory profile of the liver with samples analysed at day 8 and 14 of the culture. Data were normalised by Z-transformation to allow comparison of all analytes (white = undetected). **b** The expression of fibrosis-associated biomarkers was analysed on day 14 media samples and compared to NASH + TGFβ control samples. **c** Liver microtissues were stained for collagen-type I, α-SMA, nuclei (DAPI), phalloidin (green) and imaged by confocal microscopy. Representative images are shown and scale bars 200 μm. Staining of microtissues was quantified by measuring total fluorescence intensity throughout individual microtissues, each data point represents an average of all microtissues per MPS culture (min 8, max 20). All datapoints shown with error bars highlighting means ± SD, all data from a minimum of three independent cultures: all comparisons were made to NASH + TGFβ control samples.

survive for 3–5 days[13,14]. Multi-cell type spheroid cultures are able to recapitulate some aspects of disease biology but due to their size only have relatively small assay windows for inflammatory and fibrosis markers and are challenging to use for transcriptomic/proteomic style mechanistic studies[9–11]. The use of the automated quantitative imaging assay in this study, based on a method recently published[53] demonstrates for the first time a reproducible, robust assay to assess liver fibrosis, which is essential for the development of new therapeutic interventions for NASH. Ultimately the development of liver fibrosis, leading to cirrhosis is the histological feature that leads to irreversible damage to the liver and the loss of all liver functions[2] and is the

main predictor of liver-related mortality in NAFLD/NASH[26]. It is therefore not surprising that this is the clinical feature most targeted in NASH drug discovery, but it is also the hardest to model pre-clinically, relying on rodent models such as the use of $CCL_4$ which is known to be poorly predictive of human disease[16,54].

To complement the development of the quantitative fibrosis assay and the use of multi-plex biomarker quantification assays to assay the phenotype of the NASH MPS model, we also used RNAseq to fully detail the transcriptomic profile of the model. We compared expression profiles of the microtissues treated with isolated cues and cue combinations to markers signatures from

multiple human cohorts. These analyses demonstrated overall translatability of the model and indicated that major pathways implicated in NAFLD are expressed in the MPS NASH model. However, not every individual DEG in human NASH was recapitulated by the MPS model, possibly due to the small number of biological donors used in the study but also potentially due to that the model is limited to only the three core cell types of the liver, missing additional components supplied by the gut–liver–adipose axis. It should not be expected that any in vitro or in vivo model completely recapitulates a disease, nonetheless, when compared to commonly used rodent models of NAFL/NASH the transcriptomic profile of the MPS is far closer to the human disease. It should be noted that the data used here from the study by Teufel et al.[17] do not cover all the commonly used rodent models for NAFL/NASH and further comparative studies may be valuable.

Previous reports of MPS models demonstrate similarity to milder phenotypes of the NAFL disease spectrum[18–20] whereas in this investigation we deliberately tried to create more severe phenotypes through the application of different biological cues (fat, LPS, TGFβ, fructose, cholesterol), either individually or in combination. We observed that with the right cues we generated metabolic, inflammatory and fibrotic changes, as well as reduced hepatic function (e.g. reduced albumin production) which correlates with clinical observations for longer-term disease[55]. Not all biomarkers of human NASH were robustly altered by the cues, e.g. PARP1 overexpression was observed with fat loading, but a subsequent reduction in SIRT1 expression, matching clinical data[56] was not observed. This is likely due to some signals, particularly those from peripheral immune cells being absent from the model and additional cues might be explored in future studies to robustly affect these pathways (e.g. use of exogenous TNFα which is known to regulate PARP1[56]). An important finding was that the combination of many cues actually led to an overall suppression of gene expression and subsequent cytokine production indicating that there was significant cross-pathway interaction and also a positive association between gene expression and protein (cytokine) translation. To drive an even more severe disease phenotype (moving toward cirrhosis, with high levels of cell death) will require the presence of extra-hepatic stimuli and this should be the focus of a future investigation. Consistent with our previous findings, the presence of HSC and KC were required to drive inflammation and fibrosis but interestingly, some of the largest transcriptional changes in disease markers created in this current paradigm used what we considered to be a non-physiological number of NPCs (6K). Possible explanations for this result include the fact that with reduced NPC number there is a higher cell-cue molecule ratio, therefore, making their relative gene expression levels appear high, or that different cell ratios between in vitro and in vivo situations and may reflect a misalignment of scaling between the full organ and the miniature representation of a sinusoid. However, the production of soluble biomarkers and most importantly extracellular matrix laydown was consistently higher with greater numbers of NPCs making this form of the model most appropriate for evaluating pharmacological intervention. The treatment of the NASH microtissues with TGFβ and to a lesser extent fructose, resulted in a model with a more pronounced inflammatory and fibrosis disease state. TGFβ is a well-studied and well know regulator of fibrosis[39] and the data align closely with observations from a recent study showing how TGFβ signalling is fully recapitulated in the MPS model[21]. It should also be noted that the dose of TGFβ used in this study (1 ng/ml) is at least tenfold lower than what is used in most other in vitro models[11,57], suggesting the model is highly responsive to physiologically relevant concentrations of stimuli. Overall, the study demonstrates the

importance of an integrative approach that uses transcriptomic, proteomic, genomic, metabolomic and lipidomic data which will guide us towards building more representative models of disease development and progression.

The primary focus for any pre-clinical model such as the liver MPS described here is to be used to predict human therapeutic intervention and therefore we explored the use of two developmental compounds, OCA and ELF in this study. OCA is a semisynthetic bile acid analogue that activates the FXR to improve insulin sensitivity and can elicit anti-inflammatory and anti-fibrotic effects in the clinic[58]. ELF is a dual PPARα/δ agonist with the alpha activity increasing fatty acid oxidation while the delta activity produces anti-inflammatory effects[59]. In a phase 3 trial, OCA improved liver fibrosis while NASH resolution did not reach statistical significance in an 18-month interim analysis[60]. The final evaluation of OCA should await the completion of the REGENERATE study. Thus, the fact that OCA reduces liver fibrosis in human NASH patients fits with the reduced collagen deposition observed in the MPS NASH model. ELF (120 mg) showed a significant effect on NASH resolution in post-hoc analysis of the GOLDEN-505 phase 2b trial[59]. In a phase 3 trail, the effects of ELF on NASH resolution and fibrosis improvement did not reach statistical significance in a 16-month interim analysis[61]. The company is discussing with regulatory authorities to extend the study to obtain longer-term results. Thus, how well the MPS NASH model reflects the translation of pharmacology with ELF needs to await longer-term data from the clinical study.

The data sets generated in this study demonstrate that the MPS NASH model can be used to efficiently test new anti-NASH compounds in a little as 2 weeks, compared to months in rodent models. Exposure to each of these candidate drugs showed a clear link between inflammatory pathways and histological changes measured by confocal imaging, a finding that has been previously reported in patients[45]. Importantly, in this study we carefully scaled the in vitro dose the microtissues received so as to ensure it was comparable to what is used in vivo, which is an approach not normally taken for in vitro-based studies. We would encourage others to take this approach when using advanced in vitro models to enhance the translation of the data obtained from the platforms. Finally, it was striking that when studying fibrosis in the NASH microtissues the largest decreases in fibrotic endpoints were observed after treatment in the presence of TGFβ, demonstrating the value of being able to test the efficacy of a compound in varying disease states in the same model, potentially in parallel.

Dietary and lifestyle changes as well as bariatric surgery result in weight loss and can improve NASH symptom scores[32,33], providing proof that attenuating metabolic drivers can be an effective therapy. To this end, we investigated whether the NASH phenotype in our MPS model was equally reversible by altering the fat content of the cell culture medium. We observed that altering the medium composition halfway through a 30-day long culture of the NASH microtissues was able to halt the progression of a range of disease markers, something not previously reported for other in vitro-based approaches. This further demonstrates the utility of the MPS NASH, model for evaluating a wide range of therapeutic interventions and if combined with the transcriptomic or proteomic approaches also enables detailed mechanistic studies.

Despite being able to demonstrate that the microtissues recapitulate essential features of NASH, we do recognise that this is a complex disease and acknowledge that our model has limitations, which should be considered when interpreting data. For instance, the development of clinical NASH occurs over many years, whilst in our MPS model it occurs over several weeks, in line with typical in vivo rodent models. In addition, progression of NAFL to NASH is known to involve extra-hepatic stimuli which come

from the adipose tissue, the gastrointestinal (GI) tract, the immune system and the microbiome. Whilst not within this current model, MPS have the potential to allow tissue and/or organ models to be connected through the passing of a common medium between tissue constructs and this medium can be used to transfer peripheral immune cells[62,63]. It would be anticipated that increasing the immune component of any model would further enhance the disease severity observed in the liver microtissues. The complexity of the model could also be increased by connecting it to adipose tissue and/or the GI tract in a multi-organ MPS[63,64]. Finally, although there are monotherapies making good clinical progress for the treatment of NASH it is widely accepted that optimal treatment paradigms are likely to involve drug combinations and in particular, the drugs may not all be small molecules but may belong to the newer generation of treatments in the drug class known as biologicals. Consequently, to further demonstrate the value of the model in drug discovery and development, it would be prudent to test a broader spectrum of modalities and therapeutic approaches.

In conclusion, we have demonstrated that the NASH MPS model is able to recapitulate major features of NAFL/NASH and, using an automated imaging approach we have shown for the first time that it has a robust and quantifiable fibrosis phenotype. The transcriptional profile of the model highly correlates with that of the human disease, something which pre-clinical rodent models struggle to achieve, and we show the phenotype of the model can be manipulated through pharmacological and dietary intervention. Finally, we explored how different biological cues could further enhance the disease state in the model and demonstrated how TGFβ could be used to create and advanced NASH disease state with enhanced fibrosis and reduced hepatic function. Using the modular nature of the MPS NASH model different aspects of the disease can be modelled, allowing a wide range of studies to be performed to help identify novel therapeutic targets or test the efficacy of already developed therapies.

## Materials and methods

**Primary human cell culture and treatment**. Cryopreserved PHH, primary human KC and primary HSC were purchased from Life Technologies (Paisley, UK). Cells were thawed according to the instructions provided by the supplier. Viability was assessed using the trypan blue exclusion test and was >80% for all lots. Cultures were performed in the PhysioMimix™ LC12 MPS (CN Bio Innovations, Cambridge, UK) (Supp. Fig 1), either with PHH alone or in a co-culture of PHH with KC and/or HSC. Co-cultures were performed with two ratios of KC and HSC, either one KC/HSC per 10 PHH (high NPC) or one KC/HSC per 100 PHH (low NPC). Co-cultures were always performed with non-donor matched cells.

The MPS LC12 plates consists of 12 individual bioreactors that contain pneumatically driven micro-pumps to recirculate cell culture medium through a collagen-coated scaffold. Cells were cultured using the protocol previously described[20,21], but briefly, PHH were seeded at a density of $6 \times 10^5$ viable cells per well on the LC12 plates with each NPC population (KC and/or HSC) at a density of $6 \times 10^4$ viable cells per well. Each well contained a final volume of 1.6 mL of medium per well and maintained under a flow rate of 1.0 μL/s. Cells were seeded in Williams E medium containing primary hepatocyte thawing and plating supplements (Life Technologies, USA) and cultured initially under downward flow for 8 h before the flow was automatically reversed to upward flow for the next 16 h of culture. Microtissues were then cultured in either HEP-Lean or HEP-Fat medium, both media's are proprietary to CN Bio Innovations Ltd. The HEP-Fat medium is a derivative of the HEP-LEAN medium and contains a mixture of saturated and unsaturated FFAs, as well as physiologically relevant quantities of insulin and sugars. Complete media changes were performed on all wells every 48–72 h. For certain experiments, the media was supplemented with additional additives from day 4 onwards: 1 ng/ml LPS (lipopolysaccharides from Escherichia coli O111:B4) (Sigma Aldrich, UK), 1 ng/ml recombinant TGF beta 1 protein (Abcam, UK); 500 μM Fructose (Sigma Aldrich, UK); 50 μg/ml Cholesterol (Sigma Aldrich, UK).

**Imaging of microtissues**. To visualise fibrosis proteins in the liver microtissues, scaffolds were fixed for 15 min in 4% paraformaldehyde following culture in the MPS platform. They were permeabilized with 0.1% Triton-X and blocked with 2% BSA before staining overnight at 4 °C with antibodies against alpha SMA (Abcam, UK - ab7817) and collagen 1 (Rockland, USA—600 401 103 0.5). Scaffolds were

then stained with fluorescently conjugated secondary antibodies (Alexa-Fluor 555 goat anti-rabbit #A21429; Alexa-Fluor 647 donkey anti-rabbit #A31573; Alexa-Fluor 555 goat anti-mouse #A21424; Alexa-Fluor 647 goat anti-mouse #A21235 (Thermofisher, UK) along with Phalloidin 488 (Thermofisher, UK) and Hoechst 33342 (Invitrogen) for 1.5 h at room temperature. Scaffolds were washed three times with PBS and mounted onto glass microscope slides with anti-fade mounting medium. Confocal fluorescent images of the stained scaffolds were captured using a Cell Voyager 7000 (CV700, Yokogawa Inc.) automated confocal microscope. Image acquisition was controlled using the SearchFirst functionality of the Wako Software suite (Fujifilm Wako Automation). A workflow was designed to automatically locate and acquire images of each scaffold in a consistent and unbiased manner (Supp. Fig. 1). The workflow consisted of a first round of acquisition where the entire slide is imaged at low (4×) magnification capturing the Hoechst channel only. Images were automatically stitched together using a MATLAB script and eight regions of interest (ROI) were identified on each scaffold (same array of ROI per scaffold). XY coordinates were subsequently fed back to the microscope which subsequently imaged each ROI with full Z stacks obtained at ×10 magnification, 120 μm stack with a 5 μm slice interval. An additional MATLAB script was used to identify the Z stack image from each region with microtissues in correct focus (using Hoechst channel) and then alpha SMA and collagen-1 staining intensity were calculated for each region. These were averaged across each scaffold to give a mean alpha SMA and collagen-1 staining value.

**Biomarker profiling by Luminex and enzyme-linked immunosorbent assay**. Soluble biomarkers in cell culture medium from the MPS were analysed by enzyme-linked immunosorbent assay (ELISA); IL-6 was measured by Human IL-6 DuoSet ELISA (R&D Systems); MCP-1 was measured by Human CCL2/MCP-1 DuoSet ELISA (R&D Systems); pro-collagen 1 was measured by Human Pro-Collagen I α1 DuoSet ELISA (R&D Systems); TIMP-1 was measured by Human TIMP-1 Quantikine ELISA Kit (R&D Systems); Fibronectin was measured by Human Fibronectin DuoSet ELISA (R&D Systems); YKL-40 was measured by Human Chitinase 3-like 1 Quantikine ELISA Kit (R&D Systems); Albumin secretion was measured by Human Albumin AssayMax™ ELISA kit (Assay Pro, St. Charles, MO). For all assays, the manufacturer's protocol was followed.

Cytokine profiles in cell culture medium samples were also analysed by Luminex array for determination of 38 analytes in total, consisting of cytokines, chemokines, and growth factors using the MILLIPLEX MAP Human Cytokine/Chemokine Magnetic Bead Panel - Premixed 38 Plex (Merck Millipore, USA). In total 15 of the analytes were observed to have no detectable expression or had no significant differences between any of the culture conditions so were excluded from the analysis.

**RNA isolation and gene expression analysis**. Total RNA was extracted from liver microtissue-containing scaffolds using TRIzol® Reagent (Ambion, USA) and a chloroform phase separation. RNA was precipitated from aqueous phase samples using 100% isopropanol and RNA pellets were resuspended in dH2O. Samples were analysed by both QPCR and RNAseq. For QPCR, reverse transcription and PCR were performed using RT2 First Strand Kit and Human Fibrosis RT² Profiler™ PCR Arrays (PAHS-120Z) (Qiagen, UK), with samples analysed on a Quantstudio 6 Real-Time PCR System (Applied Biosystems, UK). Ct values from samples were compared and normalised to house-keeping gene expression. The fold-change in each transcript represented on the array plate was determined using RT2 Profiler™ PCR Array Data Analysis (v. 3.5) Web Portal (Qiagen, UK).

Prior to RNA-seq integrity and quantity of RNA were assessed by Fragment Analyzer high sensitivity RNA kit (Agilent Technologies, USA). 50 ng of total RNA was used as input to each mRNA-seq library. KAPA mRNA HyperPrep kit (Roche, Switzerland) was used for reverse transcription, generation of double-stranded cDNA and subsequent library preparation and indexing all performed according to protocol provided by manufacturer. Quality and quantity of libraries were assessed by Fragment Analyzer standard sensitivity NGS kit (Agilent Technologies, USA). Indexed libraries were pooled in equimolar ratios. The sample pools were quantified with a Qubit Fluorometer (ThermoFisher Scientific, USA) using the dsDNA HS kit (ThermoFisher Scientific, USA), further diluted and sequenced to >13 M paired reads/sample on NextSeq500 (Illumina, USA) with PE 75 base pair (bp) read length setting.

The sequencing data were demultiplexed using Illumina bcl2fastq2-v2.16. The quality of the reads was assessed using FastQC (www.bioinformatics.babraham.ac.uk/projects/fastqc). The reads were processed and mapped to the human genome using the Bcbio-nextgen framework version 1.1.2 (https://github.com/chapmanb/bcbio-nextgen). The aligner used was Hisat2 v.2.1.0[65]. The alignment quality was assessed with QualiMap v.2.2.2 (http://qualimap.bioinfo.cipf.es). Transcript-level abundances were imported and utilised for counts estimation using the TxImport package[66]. Identification of differentially expressed genes at False Discovery Rate <5% per comparison was performed using DESeq2 in R[67]. As the experiments involved 179 samples multiple batches were unavoidable. To assess batch influence, 10,000 gene comparisons (3 replicates for each gene) were simulated according to the observed batch/plate effects and the residual variability observed across the experiment. A small median per gene error rate of 4.89% was identified and was below our differential comparison p value cut-off of 5%. RNA-seq data set was deposited on GEO with accession number: GSE168285.

Differential expressed gene (DEG) lists were first compared against online databases using the Enricher gene list analysis tool (https://maayanlab.cloud/Enrichr/)[68], with comparisons made to the Human gene atlas (www.ebi.ac.uk/gxa/home) and DISEASEs database (https://diseases.jensenlab.org) for the MPS NASH model. DEG lists were also compared to published gene lists for genes over and under-expressed in NAFLD/NASH[17,30] to determine if the same directionality of expression was present in the MPS model. When cultured with and without various cues the data set was compared to other literature datasets for NAFLD progression[45], fibrosis stage progression[45], liver-specific genes[46], fibrosis markers[47], stellate cell signature[48], markers associated with HCC risk[49,50] and core metabolism and transporter genes[51].

Additionally, DEG lists from selected experimental conditions were used to identify enriched pathways and biological processes. First, the lists were filtered to include only significantly up- and down-regulated genes ($p$ value < 5%). Subsequently, these filtered lists were then uploaded on the publicly available PANTHER database[69]. Analysis conditions were set to Fisher's exact type and Bonferroni correction for multiple testing.

Secondly, the filtered lists of significantly differentially expressed genes for the various experimental conditions, were further analysed with an in-house developed R-script. In this, the code reads-in all files and filters by strength of fold-change value (e.g. +-1, +-2, +-3) to only identify the most important altered genes. A conversion from gene symbol to ENTREZID was performed via a function called mapIDs. Lastly, selected experimental conditions and their gene-expression profile were compared using a function called compareCluster, which identifies biological processes that are common between the gene sets. Eventually the top-ranking processes were graphically visualised.

**LC-MS/MS analysis for drug metabolism and protein binding**. To determine the metabolic turnover of obeticholic acid and elafibranor in the MPS platform, both compounds were incubated with liver microtissues for 48 h starting at 1 µM and media samples taken at regular intervals (AdooQ Bioscience, UK). The presence of each compound in the cell culture medium was quantified by liquid chromatography-tandem mass spectrometry (LC-MS/MS). Samples were separated using a Waters Acquity UPLC system with an ACE Excel C18-AR column and analysed on a Waters TQ-S Micro mass spectrometer against quantitative standard curves. Both compounds were also assessed for protein binding in HEP-Fat media using rapid equilibrium dialysis (RED) and quantified using the same LC-MS/MS methodology.

**Oil Red O staining and quantitation**. Scaffolds containing microtissues were removed from the MPS, washed with PBS and fixed in 4% PFA (in PBS) for 15 min. Scaffolds were washed twice with 70% isopropanol for 5 min and then stained for 60 min in Oil Red O stain, which was a 3:2 mix of Oil Red O solution (Sigma Aldrich, UK) and dH$_2$O. Tissues were washed three times in dH$_2$O and twice in 70% isopropanol to remove non-specifically bound stain. Colour bright field images of stained scaffolds were taken using an inverted light microscope (Leica, UK). Oil Red O stain was removed from tissues by incubating them with 100% isopropanol for 15 min. The level of Oil Red O in the microtissues was quantified by analysing absorbance at 515 nm. Total cellular protein was determined for each sample following staining. Each scaffold was washed once in PBS and lysed in 0.1 M NaOH + 2% SDS. Total cellular protein was measured with a Pierce BCA protein assay kit (Thermo Fisher, UK). The relative fat content of each sample was determined by normalising the level of Oil Red O, expressed as absorbance at 515 nm to the quantity of total protein.

**Exposure of cultures to anti-NASH compounds**. Liver microtissues were cultured for 4 days in HEP-FAT medium, before dosing with obeticholic acid or elafibranor. Both compounds were dissolved in DMSO, diluted in cell culture medium (final DMSO concentration ≤1%) and added to microtissues during a full medium change every two days and in addition on the intervening days compounds were spiked into the media to increase drug concentration back to desired concentrations. Cultures were maintained in the presence of the compound, for a total of 10 days in the presence or absence of TGFβ.

**Statistics and reproducibility**. All the experiments were performed with at least three replicates, with cells obtained from three different primary human hepatocyte donors, if not indicated otherwise. Values reported are means ± SD, unless otherwise stated. Comparisons between groups were performed using generally via 1-way ANOVA (with Sidak correction for multiple comparisons) or if stated a Student's t-test, both were used as two-tailed tests. $P$ values < 0.05 were considered statistically significant.

**Reporting summary**. Further information on research design is available in the Nature Research Reporting Summary linked to this article.

## Data availability statement
All transcriptomic data are deposited on GEO and is immediately available with accession number: GSE168285. The source data for the main figures are available as Supplementary Data 1 and all other data are available from the corresponding author on reasonable request.

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

## Acknowledgements

Supported by Innovate UK grant (Technology Strategy Board): Combining organ-on-chip models of fatty liver with computational systems biology to enable drug re-purposing—Ref: 103598.

## Author contributions

T. Kostrzewski, S. Snow, A. Lindström Battle, S. Peel, Z. Ahmad, J. Basak, J. Lindgren, M. Clausen and L. Young performed experiments and analysed data; M. Surakala, A. Bornot, M. Ryaboshapkina, D. Linden, C. Maass, and A. Corrigan analysed data; T. Kostrzewski, L. Ewart and D. Hughes designed and coordinated the research and wrote the paper.

## Competing interests

At the time of this study, all authors were employees of either CN Bio Innovations Ltd, AstraZeneca and Certara.
