## [Transparent Peer Review File · Communications Biology]

Reviewers' comments:

Reviewer #1 (Remarks to the Author):

In the manuscript by Kostrzewski et al. entitled 'Modeling human liver fibrosis in the context on non-alcoholic steatohepatitis using a microphysiological system' the authors introduce an in vitro microphysiological NASH model system which consists of co-cultured primary human liver cells. In this model the authors could impressively show that major features of NAFLD/NASH can be recapitulated and therapeutic interventions can be mimicked. With RNA sequencing it could be convincingly shown that the expression of inflammatory markers as well as fibrosis modulators was present in the used model system. These findings together with the ability to test pharmacological and dietary intervention is novel and of great interest for the research community. Results are presented in an accurate way and the methodology and statistic analysis is well chosen and performed. Conclusions are properly drawn and the data is critically discussed.

Only minor points of criticism arise:

1. In Figure 3H why is there no result for Fat-> Lean after Day 15?
2. Authors should comment on the known impact of infiltrating immune cells in the context of NASH progression. Can you speculate on how this could be implemented into the model system in the future?
3. The manuscript contains some minor spelling errors.

Reviewer #2 (Remarks to the Author):

The authors described human liver fibrosis using a microphysiological system. The same group published multiple articles with microphysiological system earlier and there is no aspect of novel finding at that aspect. The modeling of the liver fibrosis model is a unique approach but lack major controls. Here are few major comments:

1. It is well known that stellate cells activated in vitro in normal conditions and express fibrosis markers like α -SMA, col etc. The co-culture of hepatocytes, Kupffer cells, and stellate cells also demonstrated the same pattern. It is necessary to run individual cell culture of the same hepatocytes, Kupffer cells and stellate to demonstrate the significant difference at the level of gene expression and other markers of NASH.
2. It is well known that the PARP pathway is critical in both NAFLD and NASH models (Journal of Hepatology: 66, 3, 2017, 589-600.) The author should explain the difference in their finding with respect to the literature.
3. It is important to note that human primary hepatocytes always accumulate fat and generate lipid droplets during culture. The gene expression of fatty acid may be the sole of the contribution of hepatocytes rather than tri-coculture. Again, control experiments with single-cell type will be useful to address that question.
4. The same question is true for Kupffer cells as the author observed significant upregulation of cytokines and chemokines.
5. One of the major missing aspects is the role of the cytochrome system and mitochondria. None of those pathways are highlighted in system biology interrogation.

Minor comments:

1. Statistical analyses in figure 6b are not convincing and the authors are requested to provide a response to comments.
2. It is not stated whether three cell types were obtained from the same donor.

Point-by-point response to reviewers

	Reviewer 1 comments	Revisions and comments
1	In Figure 3H why is there no result for Fat-> Lean after Day 15?	Figure 3H adjusted to show data for Fat -> Lean at day 15.
2	Authors should comment on the known impact of infiltrating immune cells in the context of NASH progression. Can you speculate on how this could be implemented into the model system in the future?	Inflammation is a well-established driver in the initiation, maintenance and progression of NAFLD/NASH. The infiltrating of peripheral immune cells into the liver has been shown to play a role in NASH progression and in particular an imbalance between the presence of neutrophils and T cells (DOI:10.3389/fimmu.2016.00490). Currently the developed NASH model does not incorporate any peripheral immune cells but similar multi-organ MPS to the one used here have now incorporated Treg and Th17 cells to flow around microtissues. (DOI: 10.1016/j.cels.2020.02.008). It would be anticipated that the addition of a further immune component(s) would likely advance the disease state modelled owing to additional increases in cell stress. The follow comment has been added to the discussion: Line 529 – “Additionally, progression of NAFL to NASH is known to involve extra-hepatic stimuli which come from the adipose tissue, the gastrointestinal (GI) tract, the immune system and the microbiome. Whilst not within this current model, MPS have the potential to allow tissue and/or organ models to be connected through the passing of a common medium between tissue constructs and this medium can be used to transfer peripheral immune cells^{61,62}. It would be anticipated that increasing the immune component of any model would further enhance the disease severity observed in the liver microtissues.”
3	The manuscript contains some minor spelling errors.	Manuscript has been updated to remove minor spelling errors.
	Reviewer 2 comments	Revisions and comments
4	It is well known that stellate cells activated in vitro in normal conditions and express fibrosis markers like α -SMA, col etc. The co-culture of hepatocytes, Kupffer cells, and stellate cells also demonstrated the same pattern. It is necessary to run individual cell culture of the same hepatocytes, Kupffer cells and stellate to demonstrate the significant difference at the level of gene expression and other markers of NASH.	The reviewer makes an important point that hepatic stellate cells in culture in standard 2D culture will activate and express fibrotic markers. We have shown in this study that hepatocyte monocultures (labelled steatosis model) do not induce a NASH phenotype (Fig 1 and Supp. Fig 2) but do become steatotic (Supp. Fig 2D-E). These monocultures do not express any fibrotic or inflammatory biomarkers (Fig 1). The non-parenchymal cells (NPCs) in the triple co-cultures contain 90% hepatocytes and 10% KC and HSC and the set-up of the fluidic MPS with a 3D collagen-coated scaffold held in a fluidic pathway is designed for the culture of hepatocytes with and without non-parenchymal cells as has been extensively published on (DOI: 10.1074/mcp.RA117.000370; DOI: 10.1002/hep4.1450; DOI: 10.1124/dmd.116.071456; DOI: 10.1038/s41467-018-02969-8; DOI: 10.1124/dmd.116.074005). The culture of individual NPC populations is not possible in the perfused 3D MPS, as hepatocytes are required to form the bulk of the 3D microtissue as they do in vivo. The same is true for spheroid cultures and other 3D hepatic models which have been published on extensively, all of which contain hepatocytes with or without the presence of NPCs (DOI: 10.3390/cells9040964; DOI: 10.1038/s41598-019-47737-w; DOI: 10.3390/ijms20071629; DOI: 10.1371/journal.pone.0208958). To demonstrate role which the different cell types play in our MPS liver model we have included additional data in Supp. Fig 7 showing a comparison between PHH only, PHH + KC, PHH + HSC and PHH + KC + HSC co-cultures. This study demonstrates that only the tri-culture of all three cell types generates the phenotype most closely matching human NASH with production of inflammatory and fibrotic biomarkers and the greatest reductions in hepatic performance as the disease state is most progressed. We have included this data as part of our justification for the setup of the systems biology study where we have assessed the NASH microtissue with varying physiological and non-physiological levels of NPCs to allow us to determine in detail the effects of exogenous stimuli on the phenotype of the NASH microtissues. We have amended the results section to include this additional data:

		Line 242: "Here, a systems biology 'cue-signal-analysis' paradigm^{35,36} was utilized to produce large factorial datasets to enable the identification of molecular pathways of importance. We first compared monocultures of PHH with co-cultures with and without both KC and HSC to determine which combination gave the phenotype most akin to the human disease and observed that all NASH biomarkers were only produced with all three cell types present in the microtissues (Supp. Fig 7). To create a varying set ups of the NASH microtissues we created hepatocyte microtissues in the MPS with high numbers of NPC (60,000 KC and HSC – equivalent to 10% of microtissue population) and low numbers of NPCs (6,000 KC and HSC – equivalent to 1% of microtissue), representing physiological and non-physiological phenotypes cell ratios respectively. "
5	It is well known that the PARP pathway is critical in both NAFLD and NASH models (Journal of Hepatology: 66, 3, 2017, 589-600.) The author should explain the difference in their finding with respect to the literature.	Following the suggestion of the reviewer we investigated PARP1 and SIRT1 expression in our transcriptomic data. We observed a significant increase in PARP1 expression following fat challenge but not a robust reduction in SIRT1 expression. This additional data is shown in Supp. Fig 12 and described in the results: Line 278: "FFA and the combination of FFA plus cholesterol, but not the other single cues, affected lipid droplet-associated genes⁴³ in a manner resembling simple steatosis in NAFLD (Supp. Fig 9). Fructose and fat (FFA) challenge combined with fructose induced differential expression changes consistent with insulin resistance⁴⁴ (Supp. Fig. 10). Most cues also reduced the expression of mitochondrial associated genes (Supp. Fig 11) and challenge with FFA was sufficient to modulate PARP pathway, but changes were moderate and additional cues did not cause further changes (Supp. Fig 12)." We further highlight and discuss this result in the discussion: Line 435 - "Not all biomarkers of human NASH were robustly altered by the cues (e.g. PARP1 overexpression and reduction in SIRT1 expression), and additional cues might be explored in future studies to robustly affect these pathways (e.g. TNFα)."
6	It is important to note that human primary hepatocytes always accumulate fat and generate lipid droplets during culture. The gene expression of fatty acid may be the sole of the contribution of hepatocytes rather than tri-coculture. Again, control experiments with single-cell type will be useful to address that question.	The reviewer is right to highlight that primary hepatocytes through in vitro culture can accumulate lipid droplets and therefore in previous studies we established the effects of the fat in the media on primary hepatocyte mono-cultures (DOI: 10.3748/wjg.v23.i2.204) and in the same tri-coculture set up (DOI: 10.1002/hep4.1450) using lean (low sugar and low fat media as a control). We further confirmed in this study that hepatocyte monocultures cultured in the high fat media (labelled steatosis in Fig 1 and Supp. Fig 2) do not generate a significant NASH phenotype (as assessed by inflammatory and fibrotic endpoints) but do become steatotic and microtissues are loaded with fat (Supp. Fig 2D-E). In addition to this further data has been provided in Supp. Fig 7 (as discussed in detail in point #4) to show the NASH phenotype of the liver MPS with and without each cell type.
7	The same question is true for Kupffer cells as the author observed significant upregulation of cytokines and chemokines.	This is a follow-on question from question 4 and question 6. We show in the new additional data presented in Supp. Fig 7 that PHH and KC co-cultures in the MPS set up do not produce inflammatory or fibrotic biomarkers. Additional previous publications (DOI: 10.1038/s41467-018-02969-8 and DOI: 10.1124/dmd.115.063495) have shown that PHH and KC co-cultures do not produce inflammatory cytokines and chemokines without the presence of significant exogenous stimuli (e.g. viral infection or TLR agonist activation). The data presented shows that the presence of the three cell types in the NASH microtissues induce some level of crosstalk induce cytokine production from both KC and HSC cells.
8	One of the major missing aspects is the role of the cytochrome system and mitochondria. None of those pathways are highlighted in system biology interrogation.	The authors would like to direct the reviewer to Supp Fig. 16 where a detailed analysis of cytochrome (phase 1 enzymes) gene expression was performed as part of the system biology study, this is also performed alongside an analysis of phase 2 metabolic enzymes and liver transporters. All of which show the considerable effect of TGFβ and combinations of cues incorporating TGFβ which reduce expression of all the groups of ADME liver genes. The reviewer however is right to highlight the lack of analysis of mitochondrial genes, we did not emphasize mitochondrial function in the manuscript as our main focus was fibrosis. We looked up GO term enrichment scores for mitochondrial function from systematic analysis and an additional supplemental figure has been generated to show

		the effect of the biology cues on these genes (see Supp. Fig 11) which are generally reduced as the disease progresses in the MPS mimicking human data. The results section text has also been updated to reflect this addition: Line 274: “We firstly determined if the microtissues recapitulated the molecular signatures of processes that occur early in the development of NAFLD in humans, namely lipid accumulation, insulin resistance, mitochondrial dysfunction and changes to the poly (ADP-ribose) polymerases (PARP) pathway. FFA and the combination of FFA plus cholesterol, but not the other single cues, affected lipid droplet-associated genes⁴³ in a manner resembling simple steatosis in NAFLD (Supp. Fig 9). Fructose and fat (FFA) challenge combined with fructose induced differential expression changes consistent with insulin resistance⁴⁴ (Supp. Fig. 10). Most cues also reduced the expression of mitochondrial associated genes (Supp. Fig 11) and challenge with FFA was sufficient to modulate PARP pathway, but changes were moderate and additional cues did not cause further changes (Supp. Fig 12). Interestingly, there were differential gene expression changes with the expected directionality to NAFLD in the microtissues with the lower number of HSCs and KCs (6K) as well as the microtissues with a more physiological number (60K) of non-parenchymal cells (Supp. Fig 9-10) and the changes in mitochondrial genes were most pronounced in the low NPC group (Supp. Fig 11).”
9	Statistical analyses in figure 6b are not convincing and the authors are requested to provide a response to comments.	The statistical analysis in Fig 6b shows comparisons between the NASH + TGFβ dosed control all other conditions by 1-way ANOVA. Statistical significance for these comparisons has been checked and the figure panel updated. It should be noted that in this study all the compound dosed samples are on a NASH + TGFβ background so all comparisons are made to that control not the NASH only control, which is shown for reference. The figure legend has also been updated to make this clear and reads: “Throughout the compound dosing period TGFβ was also dosed onto all microtissues (NASH only vehicle control samples were also included in the study for comparison, which were not treated with TGFβ or dosed with any compound). A) The secreted cytokine profile from treated liver microtissues was compared by Luminex to determine the effects of each compound on the inflammatory profile of the liver with samples analysed at day 8 and 14 of the culture. Data were normalised by Z-transformation to allow comparison of all analytes (white = undetected). B) The expression of fibrosis-associated biomarkers was analysed on day 14 media samples and compared to NASH + TGFβ control samples.”
10	It is not stated whether three cell types were obtained from the same donor.	The three cell types used in all the studies were never obtained from the same donor. A line has been included in the results and the materials and methods to state this: Line 152 – “We additionally established that the phenotypic changes observed in the NASH microtissues were not affected by the biological donors used, particularly as non-donor matched material was always used to generate NASH microtissues (Supp. Fig 3).” Line 565 – “Co-cultures were always performed with non-donor matched cells.”
Revisions to align with editorial policies		
	Revisions to align with editorial policies	Revision made
11	Plotting of individual data points on all graphs	Figure 3 - individual data points plotted on graphs in panels B, E, F, G and H
12		Supplemental Fig 2 - individual data points plotted on graphs in panel A and E
13		Supplemental Fig 3 - individual data points plotted on graphs in panel B and C
14		Supplemental Fig 6 - individual data points plotted on graphs in panel B and C
15		Supplemental Fig 16 – individual data points plotted on graphs in panel B and C
16	Tables have been updated to remove shading	Table 1 – shading removed
17		Supplemental Table 1 – shading removed
18		Supplemental Table 2 – shading removed
19	Abstract reduced in length	Abstract length reduced closer to the desired length of 150 words

REVIEWERS' COMMENTS:

Reviewer #2 (Remarks to the Author):

The authors convincingly addressed all reviewer comments.

Reviewer #3 (Remarks to the Author):

The authors tried their best to address all concerns. However, I do not agree with them that individual hepatocytes or NPC can not be grown in 3D (or 2D should be work great as control). Being said that, the authors did a great work overall.

The major question I have now is the follow up experiment with PARP finding. The authors found that PARP1 gene are induced but no alteration in SIRT1. I would suggest to perform at least PARP activity. The lack of SIRT1 modulation is understandable in the context that this MPS model of NASH do not have any liver injury phenotype. PARP activity will support it further. Alternatively, the authors must describe this limitation with relevant literature as provided in the first comment.

There are numerous typo and error in the manuscript particularly in the supporting materials, which must be corrected.

Point-by-point response to reviewers

	Reviewer 3 comments	Revisions and comments
1	The lack of SIRT1 modulation is understandable in the context that this MPS model of NASH do not have any liver injury phenotype. PARP activity will support it further. Alternatively, the authors must describe this limitation with relevant literature as provided in the first comment.	We have added some additional comments in the manuscript discussion to address this point and reference the literature suggested by the reviewer. Line 423 – “Not all biomarkers of human NASH were robustly altered by the cues, e.g. PARP1 overexpression was observed with fat loading, but a subsequent reduction in SIRT1 expression, matching clinical data ⁵⁶ was not observed. This is likely due to some signals, particularly those from peripheral immune cells being absent from the model and additional cues might be explored in future studies to robustly affect these pathways (e.g. use of exogenous TNF α which is known to regulate PARP1 ⁵⁶).
2	The manuscript contains some minor spelling errors particularly in supporting materials.	Manuscript has been updated to remove minor spelling errors and to ensure all details are articulated, particularly in supplementary materials.
	Revisions to align with editorial policies	Revision made
3	Exact p-values added to graphs	Exact p-values added to all main figures where appropriate
4	Define error bars	All error bars defined in figure legends
5	Scales bars required on all microscope images	Scale bars added to panels in Fig 3, Fig 6 and Supp Fig 1
6	Avoid use of “as described previously”	Additional details added to section: Primary human cell culture and treatment
7	All details on use of antibodies described	Additional details added to methods section: Imaging of microtissues
8	Data availability statement	Statement added to the end of the methods statement